# A Multilayer Perceptron-Based Spherical Visual Compass Using Global Features

**DOI:** 10.3390/s24072246

**Published:** 2024-03-31

**Authors:** Yao Du, Carlos Mateo, Omar Tahri

**Affiliations:** 1Université Bourgogne, 21000 Dijon, France; yao.du@insa-cvl.fr; 2ICB UMR CNRS 6303, Université Bourgogne, 21000 Dijon, France

**Keywords:** omnidirectional cameras, robots, machine learning, global feature extraction, robot vision systems, localization

## Abstract

This paper presents a visual compass method utilizing global features, specifically spherical moments. One of the primary challenges faced by photometric methods employing global features is the variation in the image caused by the appearance and disappearance of regions within the camera’s field of view as it moves. Additionally, modeling the impact of translational motion on the values of global features poses a significant challenge, as it is dependent on scene depths, particularly for non-planar scenes. To address these issues, this paper combines the utilization of image masks to mitigate abrupt changes in global feature values and the application of neural networks to tackle the modeling challenge posed by translational motion. By employing masks at various locations within the image, multiple estimations of rotation corresponding to the motion of each selected region can be obtained. Our contribution lies in offering a rapid method for implementing numerous masks on the image with real-time inference speed, rendering it suitable for embedded robot applications. Extensive experiments have been conducted on both real-world and synthetic datasets generated using Blender. The results obtained validate the accuracy, robustness, and real-time performance of the proposed method compared to a state-of-the-art method.

## 1. Introduction

The task of a visual compass is to determine the direction of a camera from images. The application of visual compass can be classified into different types based on various criteria, including type of used sensor, method of estimation, and type of used features. Depending on the used sensors, visual compasses can be classified as monocular cameras [1,2,3], stereo cameras [4,5,6], RGB-D camera [7,8,9], polarized light camera [10,11,12], omnidirectional cameras [3,13,14], or a combination of sensors [15,16]. Meanwhile, visual compasses have different estimation methods by focusing on various types of information, such as 2D-3D correspondence [17,18,19], virtual visual servoing (VVS) [20,21,22], holistic method [23,24,25], extended Kalman filter (EKF) [26,27,28], and appearance-based method [3,13,29].

Methods based on 2D-3D correspondence need to match geometrical features in the spherical image. However, accurate 2D-3D correspondence on spherical surfaces is not easy under the presence of noise, occlusions, deformations, lack of textures, etc. This can lead to matching errors, affecting the accuracy of the compass.

Holistic methods attempt to utilize spherical images directly from compass estimation without relying on specific feature points or correspondences. However, the computational complexity of the overall method is high due to the transformation and shape geometrical complexity of spherical images. Extended Kalman filtering is sensitive to the choice of initial pose and the dynamic changes of the pose. In addition, appearance-based methods are also sensitive to lighting and texture changes, which leads to instability in pose estimation. Visual compass using spherical cameras is a relatively mature research topic in the field of visual navigation for mobile robotics and autonomous systems [14]. Spherical images can be obtained from a wide range of sensors such as conventional, catadioptric, fisheye, and Ricoh360 (forming a complete spherical panorama by combining images from two fisheye lenses). This paper falls into the same category of works and proposes a method for a visual compass combining formal and learning-based methods, using a wide-field-of-view camera. In the next section, the related works and the contribution of the papers are recalled and explained. In Section 3, the used methodology is explained. Section 4 provides the validation of the results.

## 2. Related Work and Contributions

A visual compass is a critical tool for mobile robot and autonomous navigation [30], and it has been widely used in various aspects [4,14,30,31]. Unlike methods such as six-degrees-of-freedom estimation or visual gyroscope, the visual compass not only proves advantageous in specialized applications but also improves robustness, making it more resilient to environmental challenges. Furthermore, its computational efficiency sets it apart, enabling real-time performance crucial for dynamic scenarios.

The field of research on the visual compass based on spherical images is an active and promising area. The work presented by Huang et al. [32] uses fully spherical images. Instead, our approaches uses just a half sphere which helps to simplify calculations. The method presented in [17] determines the camera pose from 2D to 3D corner correspondence by recovering edge-line directions, computing an orientation matrix, and determining the location vector. The work [18] introduces a real-time 2D-3D correspondence method utilizing depth information. Yoli and Ron [33] use DNN models to estimate both translational and rotational motion. While Kim et al. [34] propose a method combining learning methods and uniformization of distorted optical flow fields. Both cases can be considered as complementary solutions to our work.

The appearance-guided monocular omnidirectional visual odometry proposed in [13] utilizes appearance information. While [29] estimates position and orientation using an appearance manifold. The work [26] proposes an EKF Estimation using dual quaternion, whereas [27] implements EKF estimation based on recognized objects captured by the camera. The multi-camera pose estimation is addressed in [20] using centralized and decentralized fusion structures. Markerless pose estimation and tracking for robot manipulators are proposed in [21]. Furthermore, Horst and Moller in [35] employ spherical panoramic images for robot orientation estimation. In addition, Andre et al. [36] present the Direct Visual Gyroscope method based on dual-fisheye cameras for precise orientation estimation in applications such as robotic motion estimation and video stabilization. Holistic descriptors are introduced in [23], and their performance is tested for position and orientation estimation. Fleer et al. [24] compare holistic and feature-based visual methods. Unlike methods based on local features, holistic methods, such as methods using spherical harmonic [37], Fast Fourier transform [38] and spherical moments [39,40], can use different filters or masks to select informative regions with different qualities. Additionally, the use of masks allows us to reduce the effect of the border, especially for a catadioptric camera because assigning different weights to different regions in the image contributes to a reduction in noise and changes in lighting condition interference.

This paper proposes an efficient method for using masks to select different regions in images and utilization of the Multilayer Perceptron (MLP) to improve robustness and accuracy. On the same research line, Fekri et al. [41] introduce a tuned three-layer perceptron utilizing features from pretrained CNNs for more accurate cervical cancer diagnosis, showcasing the efficacy of combining classical MLPs with deep learning methods. New approaches such as that proposed by Yu et al. [42] have emerged, presenting a novel methodology exclusively utilizing channel-mixing MLPs and introducing a spatial-shift operation for inter-patch communication. Tolstikhin et al. [43] present an architecture based solely on multi-layer perceptrons (MLPs) for achieving competitive scores on image classification benchmarks. They demonstrate similar performance to attention-based networks [44] without requiring them. In addition to introducing the new method proposed in this paper, a new synthetic dataset has been generated using Blender. The urban canyon environment [45] and a custom omnidirectional camera model from [46] were utilized. In contrast to previous studies, we incorporated simulated weather conditions to evaluate the robustness of the visual compass in challenging environments, including foggy weather and harsh lighting conditions. The accuracy and robustness of the visual compass were thoroughly assessed under these conditions.

## 3. Method

### 3.1. Spherical Moment and Rotation Covariant Triplets

Spherical images can be obtained through various sensors and projection models. For example, the unified central model [47] can be employed to generate a spherical image from fisheye or catadioptric images. Additionally, other models treating lenses as a compact imaging system and employing a polynomial fitting model are also viable options. In this paper, the unified camera model was employed to generate spherical images from the real dataset, while a polynomial fitting model was utilized for the simulated data.

From spherical images, the spherical moment can be computed as follows,
(1)$$m_{ijk}=\int_{s}x_{s}^{i}y_{s}^{j}z_{s}^{k}I\left(s\right)ds,$$
where $m_{ijk}$ is the spherical moment of order p=i+j+k, integrating the corresponding gray-level pixels in image *I* over the spherical surface *s*. A prior study [48] has demonstrated the potential of utilizing a global feature known as triplets, which are extracted from spherical moments, to compute the camera’s rotational motion in closed form.

An example of moment triplets is shown in the following equation,
(2)$$x_{v}\;=\;\left[\begin{array}{c} m_{003}m_{101}+m_{012}m_{110}+m_{021}m_{101}+m_{030}m_{110}+m_{101}m_{201}+m_{102}m_{200}+m_{110}m_{210}\\ \;+m_{120}m_{200}+m_{200}m_{300}\\ m_{003}m_{011}+m_{011}m_{021}+m_{012}m_{020}+m_{020}m_{030}+m_{011}m_{201}+m_{102}m_{110}+m_{020}m_{210}\\ \;+m_{110}m_{120}+m_{110}m_{300}\\ m_{002}m_{003}+m_{002}m_{021}+m_{011}m_{012}+m_{011}m_{030}+m_{002}m_{201}+m_{101}m_{102}+m_{011}m_{210}\\ \;+m_{101}m_{120}+m_{101}m_{300} \end{array}\right]$$

The triplet $x_{v}$ behaves similarly to a set of three-dimensional points regarding rotational motions. This implies that after a rotational motion defined by a rotation matrix R, $x_{v}$ will undergo the following transformation,
(3)$$x_{v}^{\prime }=Rx_{v}$$

The 3D rotation matrix can be obtained from at least two non aligned triplets. From more than two triplets, the Procrustes method [49] can be used as in [48].

### 3.2. Proposed Method

The idea of using spherical moment triplets to estimate rotational motions was first introduced in [50]. As previously noted, a significant obstacle in employing global photometric features is the alteration in the image caused by the emergence and disappearance of elements within the scene from the camera’s field of view as it moves. To mitigate the abrupt variations caused by this issue, previous studies have employed weights in the form of exponential functions, as seen in [51], to develop a novel visual servoing scheme. Using a weighting function on the image has led to a better convergence domain for the servoing task.

In this paper, the same idea is used, but this time simultaneous image masks are used to obtain multiple estimated rotations. The pipeline of the proposed method is depicted in Figure 1. Consequently, a shallow network is employed to amalgamate these estimations, considering the nonlinearities arising from translational motions or the weighting functions themselves, among other factors. Nonetheless, implementing hundreds of exponential function weights, as in [51], is time-consuming and would result in non-real-time methods. The next section proposes a new way of using masks to select different regions of the image and compute the spherical moments in an effective way, while keeping real-time constraint.

### 3.3. Polynomial Mask for Selecting Specific Image Region

Considering the specific application, a well-adapted mask is the one shaped like a ring, maintaining a flat profile around the center of the region and gradually decreasing towards the mask border. A suitable candidate that holds these conditions is defined as,
(4)$$\left\{\begin{array}{cc} W\left(z_{s}\right)=e^{-\frac{{\left(z_{s}-\left(z_{0}-r\right)\right)}^{2}}{\sigma^{2}}} & \left(-1<z_{s}<z_{0}-r\right)\\ W(z_{s})=1 & \left(z_{0}-r\leq z_{s}\leq z_{0}+r\right)\\ W\left(z_{s}\right)=e^{-\frac{{\left(z_{s}-\left(z_{0}+r\right)\right)}^{2}}{\sigma^{2}}} & \left(z_{0}+r<z_{s}<1\right) \end{array}\right.$$The mask’s shape is determined by the σ and range *r* parameters, while its center is determined by $z_{0}$. Such a shape is well adapted to recover the rotation around the *z*-axis. However, its formulation is quite complex for computing moments corresponding to the selected region. Instead of using Equation (Equation 4), masks under a polynomial form on the variate $z_{s}$ can be employed,
(5)$$p_{m}\left(z_{s}\right)=\sum_{l=0}^{n}a_{l}\cdot z_{s}^{l}$$
where $a_{l}$ is the coefficients of the polynomial. They are defined such that Equation (Equation 5) approximates the shape of the mask defined by Equation (Equation 4). As an example, Figure 2c shows a mask $W(z_{s})$ corresponding to the parameters $z_{0}=0$, r=0.1, and σ=0.3 and two polynomial masks of orders 8 and 12 that fit it. Meanwhile, Figure 2d–f show the spherical images after applying rings masks centered on different $z_{0}$. In the next section, fast moment computation using a polynomial mask from the original image moments is explained.

The diagram of the proposed method is depicted in Figure 3.

### 3.4. Spherical Moment after Applying Polynomial Mask

In general, the spherical moment of a masked image is formulated as follows,
(6)$$m_{ijk}^{mask}=\int_{s}x_{s}^{i}y_{s}^{j}z_{s}^{k}I_{mask}\left(s\right)ds,$$
where $I_{mask}\left(s\right)=I\left(s\right)p_{m}\left(z_{s}\right)$. Using the polynomial formula of the mask (Equation 6), we obtain the following formulation,
(7)$$m_{ijk}^{mask}=\int_{s}x_{s}^{i}y_{s}^{j}z_{s}^{k}\sum_{l=0}^{n}a_{l}\cdot z_{s}^{l}I\left(s\right)ds.$$After a few calculation steps, we obtain that
(8)$$m_{ijk}^{mask}=\sum_{l=0}^{n}a_{l}\cdot m_{i\;j\;k+l}$$This means that the spherical moment of the image with the mask is a linear combination of the higher-order spherical moments of the original image defined by the coefficients of the mask polynomial. Using Equation (Equation 8) to compute moments of an image with a single mask requires $10^{-5}$ s with Matlab R2023b. As an outcome, a large number of masks can be used while holding real-time requirements. In contrast, under the same experimental conditions, the calculation of moments directly on 512 × 512 image using Equation (Equation 1) would take 40 ms.

### 3.5. Neural Network to Estimate the Rotation Angle

In this section, we present the used Multi-Layer Perceptron MLP architecture for refining the estimated rotation obtained through the Procrustes method, as discussed previously. The utilization of MLP offers several advantages in our approach. Firstly, it improves the accuracy of rotation estimation by performing a weighted average of each estimated rotation associated with individual masks. This improves the accuracy of the overall estimation by incorporating information from multiple sources. Secondly, our MLP is used for modeling the non-linear relationships between the presented motion inside the spherical images and the motion estimation. This aspect is particularly crucial for rotation estimation when there is translation involved in the camera motion.

For optimizing the MLP, we employed a grid search method to tune the hyperparameters. In our case, the set of hyperparameter values—depths (number of layers) and widths (number of neurons per layer)—were systematically tested in combination, forming a grid of possible configurations. The algorithm then evaluates the performance of the neural network for each combination of hyper-parameter values on a predefined metric (here the mean squared error).

#### 3.5.1. Shift Folder–Window Data Augmentation

To better leverage the sequential information embedded in the acquired data, we employ a strategy that involves shuffling data within a mobile time-window across each *n*-folder. The basic principle of this strategy is to split each sequence into *n* uniform folds and then shift a time-window as shown in Figure 4. This results in the generation of a set of image pairs within each time-window.

The initial step involves the generation of *n*-folders according to the sequence and folder sizes. Specifically, the number of folders is determined by dividing the total number of images by the predetermined folder size. Subsequently, a time-window of dimension *w* is established, and a sliding window approach is employed to systematically obtain all feasible pairs of images.

For example, with a window size of w=5, there are only four pairs of images: $\left[I_{i+0},I_{i+1}\right]$, …, $\left[I_{i+3},I_{i+4}\right]$. However, following the inner shuffle strategy proposed here, this count rises to 20 pairs, encompassing variations like quicker motion and the reversal of motion direction. These pairs of images are obtained by all the possible combinations with repetitions: $\left[I_{i+0},I_{i+1}\right]$, …, $\left[I_{i+0},I_{i+4}\right]$, $\left[I_{i+1},I_{i+0}\right]$, …, $\left[I_{i+2},I_{i+0}\right]$, …, $\left[I_{i+2},I_{i+4}\right]$, …, $\left[I_{i+4},I_{i+3}\right]$. The number of generated pairs of images, *k*, is calculated as
$$k=w\left(w-1\right).$$This data augmentation technique enables training the model using images captured at diverse intervals and directions, thus encompassing a range of different rotational speeds. Moreover, it can help to better generalize to new, unseen data by creating transformed versions of existing ones.

#### 3.5.2. Network Architecture

We have proposed the use of MLP-based architecture to refine the estimated rotation in the early stages. This MLP is a shallow net with an small number of neurons, between 16 and 128. In fact, MLPs with fewer layers generally require less computational resources, making them more efficient for training and inference. This is crucial, especially in scenarios where computational power is limited, such as in embedded systems or real-time applications. Models with fewer layers consume less memory, making them more suitable for deployment on resource-constrained devices. This is particularly important also for embedded systems, like those equipped in Unmanned Aerial Vehicles UAV, where memory limitations are common. In addition, smaller MLP architectures are less prone to overfitting, especially when the size of the dataset is limited, like in our experiments. Limiting the number of layers helps to prevent the model from capturing too many unnecessary details from the training set. Simpler models with fewer layers are often more interpretable. This can be crucial in certain applications, such as healthcare or autonomous driving, where model transparency is essential.

## 4. Experiments

The evaluation was carried out with an Intel Xeon Gold 5518 CPU @ 2.30 GHz (Hillsboro, OR, USA) PC equipped with 128 GB of memory RAM and a NVIDIA GPU A5000 (Taiwan) with 24 GB of memory. In order to generalize the results and augment the available data, we used both synthetic and real images. Specifically, for real images, we used the PanoraMis dataset [52], while we have generated synthetic images using Blender software and the canyon world [46]. The *n*-folders have been generated using a size of folder f=50 for each one of the sequences: sequences 5 and 6 (real-image dataset) and two canyon sequences (virtual-image dataset). In addition, each folder uses 80% of the images for training and 20% for validation. The time-window size in this experiment was set to w=3. To further elucidate the experimental procedures and results, supplementary video footage (Appendix A) has been provided, offering a comprehensive visual representation of the conducted experiments.

After applying the grid search strategy, we find that a single layer with a width of 16 neurons is enough for this task. This suggests that simpler architectures, specifically a one-layer network are enough to address the problems presented in our task. The Adam algorithm was used as an optimizer with the mean squared error MSE distance as a loss function. In order to accelerate the model’s learning process and prevent it from getting stuck in local minima, firstly the learning rate was set to 0.0001, then a combination of a decaying learning rate strategy and Stochastic Weight Averaging (SWA) learning rate schedule was utilized.

### 4.1. Data Preparation

The real-world data from Sequence 5 of PanoraMis consist on a sequence of 156 images of pure rotational motion without translation, while data from Sequence 6 of PanoraMis involve a combination of rotational and translational motion in a sequence of 318 images. Our own synthetic dataset from canyon world also comprises a mixture of rotational and translational motion, with the trajectories being identical across datasets, with a total of 2318 images. The only distinction lies in the simulation conditions, where one dataset replicates adverse weather conditions (poor lighting and heavy fog), while the other simulates good weather conditions. Examples of images from the data set are shown in Figure 5. Thus, our dataset is composed of a total of 2792 images, where 17% of them are real and the other 83% are simulated images. Because we use a spherical lens in both cases, real and virtual worlds, the camera frame (*z*-axes) is pointing up to keep all of the scene always visible.

The rest of the section will be divided in two main parts. In the first part, experiments are conducted to optimize the masks parameters, especially the range *r*. Then, in the second part, the proposed method is compared to a method from the state of the art [53].

In order to study the influence of the mask range on the estimation, 10 different values of *r* have been selected. After multiple tests, we found that using 50 masks for each image resulted in the minimum error for each value, and consequently, we have decided to consistently use 50 masks for each image in subsequent experiments. In terms of error calculation, we employed the Relative Pose Error (RPE). This means that a smaller error indicates a more accurate estimation of results. The mask centers $z_{0}$ are positioned uniformly between z=−0.75 and z=0.25 with σ equal to 0.2 (best parameter obtained through several trials). Moreover, for each *r*, different models were trained, respectively, and tested separately. Besides the image, its gradient norm was also used as an input to compute the moment triplets. The results of experiments on real-world datasets are presented in Figure 6a,b, while synthetic datasets are depicted in Figure 6c,d. From the 4 plots of Figure 6, it can be seen that in the case where 50 masks are used, the best results have been obtained for r=0.2 and r=0.1, where the error of *R* and ΔR obtained using the image and image gradient.

### 4.2. Comparison with an Approach from the State of the Art

In this part, first, the local feature extractors (such as FAST, ORB, and SURF) have been tested using some part of the synthetic data-set. As is shown in Figure 7, the feature extractions can fail to produce sufficient and stable features. Actually, failure in the feature extraction induces the failure of the methods based on points as features.

In the second part, the proposed methodology involving Multilayer Perceptron (MLP) combined with triplets is evaluated against the approach presented in Morbidi and Caron [53] across four distinct datasets. During the experimental phase, a mask range of r=0.2 was applied, and both raw and gradient images were utilized as inputs for the MLP/triplets’ pipeline.

Figure 8a illustrates the estimated $\theta_{z}$ values for Sequence 5 (PanoraMis) obtained using MLP/triplets and Morbidi and Caron’s method. The results reveal a similar performance across all methods, indicative of consistent outcomes. This uniformity is anticipated due to the predominantly rotational motion exhibited in Sequence 5. The same conclusion can be made for the estimated rotation between two consecutive camera positions Figure 9a, where linear motion is negligible. Figure 10a shows the error comparison of three methods. The errors associated with all methodologies are observed to be below 0.1 radians. Our method, both variations $MLP_{I}$ and $MLP_{\Delta I}$, has a large error before 100 frames, while the Morbidi and Caron method shows a large cumulative error after 100 frames. The video attached to this paper shows a better tracking of the rotation using the MLP/triplet and the image gradient as visual information.

Figure 8b shows the results of the experiments using Sequence 6 (PanoraMis). The data processed by the MLP algorithm exhibit a smoother and more precise motion, aligning almost seamlessly with the ground truth. In addition, in the comparison between raw and gradient images, we can see that the method using gradient images has a slight advantage. The same conclusion as Figure 8b can be drawn in Figure 9b. Compared with Morbidi’s method, the method using MLP/triplets proposed in this paper has advantages in the mixed motion of rotation and translation. This becomes more obvious from Figure 10b. It is evident that the method employing Multilayer Perceptron (MLP) exhibits superior accuracy, particularly as errors accumulate over time, thereby accentuating this advantage.

Similar to the real-world dataset, when employing the synthetic dataset under favorable conditions, Morbidi’s method demonstrates effectiveness primarily when the translational component of motion is negligible in comparison to the rotational component. As shown in Figure 8c, due to the effect of translation, Morbidi’s method does not work properly. As can be seen clearly in Figure 9c, the relative rotation between two consecutive images was not estimated correctly as well. In contrast, MLP/triplets provides better estimation (refer to Figure 8c), with smaller accumulated errors (refer to Figure 10c), especially in the last 400 images. MLP’s effectiveness in estimation relative rotational motion is evident in Figure 9c, where it provides a smoother and more close to the ground truth values.

In the final experiment, synthetic data were utilized under adverse conditions. Figure 8d reveals that Morbidi’s method continues to face challenges due to translation effects. Conversely, the performance of the MLP/triplets remains consistent with that observed in the previous experiment. Figure 9d exhibits similarities to the prior experiment, with notable oscillations observed around the 100th frame. This variability may be attributed to environmental interference under harsh conditions, resulting in fluctuations in grayscale values. Regarding error estimation, as illustrated in Figure 10d, the method utilizing MLP/triplets yields higher maximum errors compared to ideal lighting conditions.

## 5. Conclusions

This paper introduces a method for visual compass that combines a Procrustes solution with a shallow neural network, utilizing a panoramic camera (such as a fisheye or catadioptric camera). Polynomial masks are employed to address various image regions while adhering to real-time constraints. Additionally, our experiments have demonstrated that beyond utilizing a single layer with 16 neurons, further complexity in terms of either width or depth did not yield significant performance improvements. The adoption of lightweight Multilayer Perceptrons (MLPs) ensures faster processing times while retaining the ability to capture patterns and features, thus maintaining runtime efficiency. This is particularly important considering the reliance on the Procrustes method, which incurs residual computational costs, as well as the computational cost associated with computing moments.

To evaluate the performance of the visual compass under challenging conditions, a new dataset is generated. Comparative analysis with feature-based methods using this harsh-condition dataset reveals that the method proposed in this paper exhibits superior robustness and accuracy. Future research efforts will be directed towards employing similar schemes for translation estimation.

## Figures and Tables

**Figure 1 sensors-24-02246-f001:**
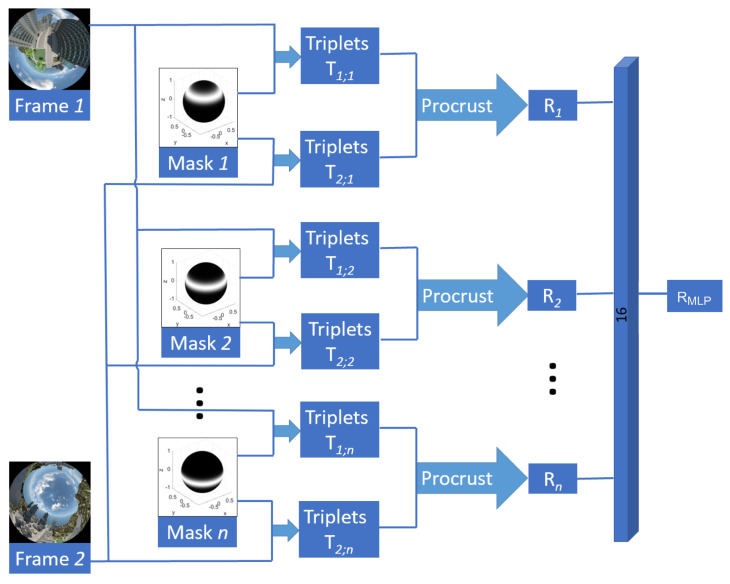
Pipeline of the proposed method. It takes as input a pair of images with a set of *n* masks and generates as output a rotation transformation $R_{\mathrm{MLP}}\in SO(2)$.

**Figure 2 sensors-24-02246-f002:**
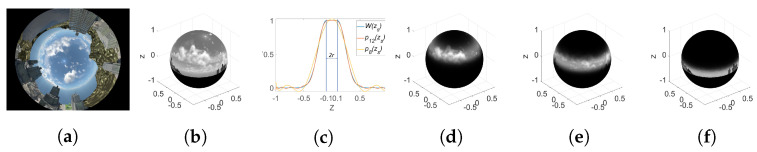
Mask shape and projection onto spherical image. (**a**) Original image. (**b**) Projected image onto sphere. (**c**) Polynomial mask shapes. (**d**–**f**) Image with mask centered, respectively, on $z_{0}=0.6$, $z_{0}=0.1$, $z_{0}=-0.2$.

**Figure 3 sensors-24-02246-f003:**
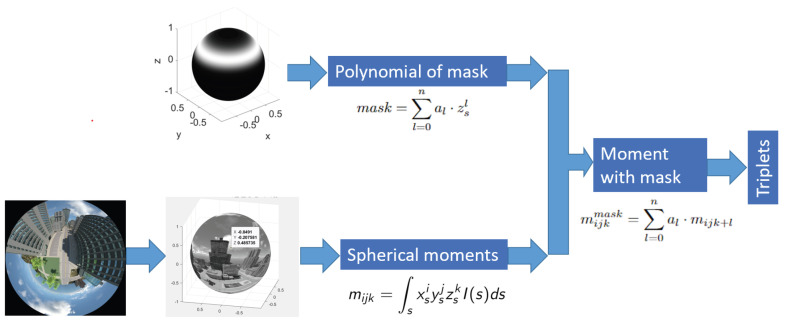
Triplet computation framework.

**Figure 4 sensors-24-02246-f004:**
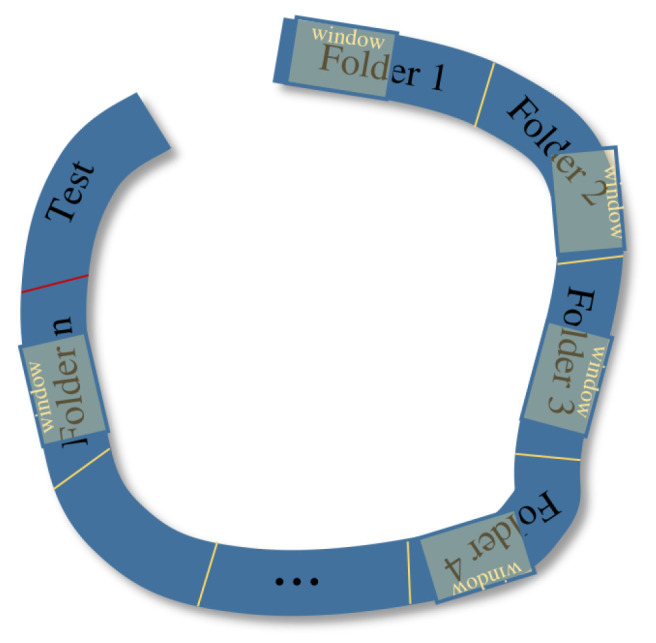
Shift folder–Window data augmentation. The thick blue curves represent the movement trajectories of the robot, which are divided into *n* folders. The yellow rectangles are the window, they slide in their respective folders, and the images located in the windows are reassembled. Red line split a segment of the trajectory to isolate test images from the training parts segments (folders).

**Figure 5 sensors-24-02246-f005:**
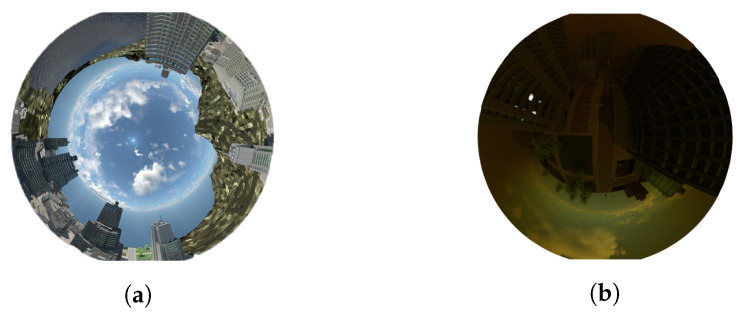
Examples of used synthesis images in good and in harsh conditions obtained using Blender 2.76b. (**a**) Synthetic image. (**b**) In harsh environments.

**Figure 6 sensors-24-02246-f006:**
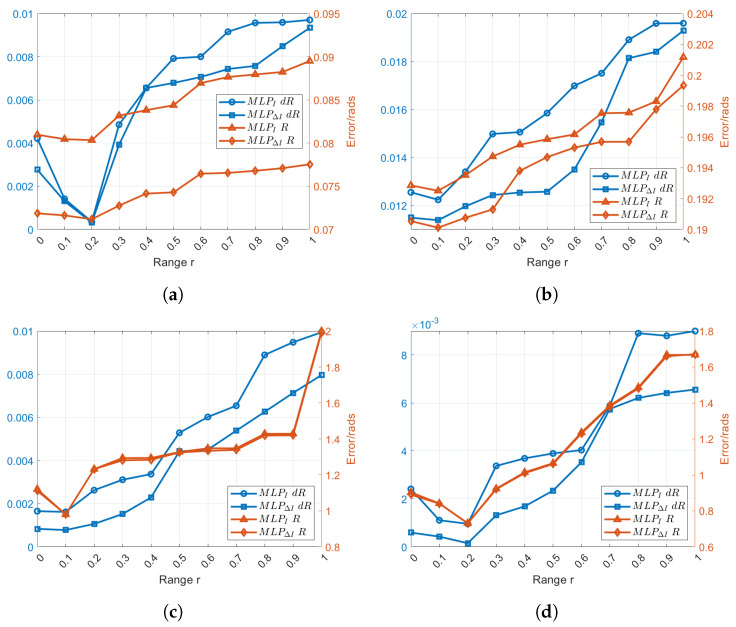
Comparison of error in various ranges *r*. (**a**) Real-world datasets from PanoraMis sequence 5. (**b**) Real-world datasets from PanoraMis sequence 6. (**c**) Synthetic datasets with general lighting conditions. (**d**) Synthetic datasets with Harsh conditions.

**Figure 7 sensors-24-02246-f007:**
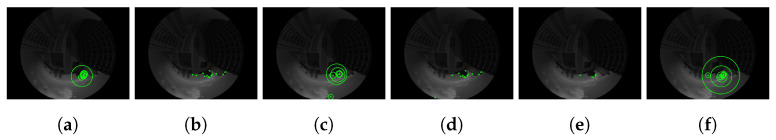
The figure above shows the 20 strongest feature points, represented by green dots and circles, obtained using 6 different feature point extraction algorithms. (**a**) ORB. (**b**) MinEigen. (**c**) SURF. (**d**) Harris. (**e**) FAST. (**f**) BRISK.

**Figure 8 sensors-24-02246-f008:**
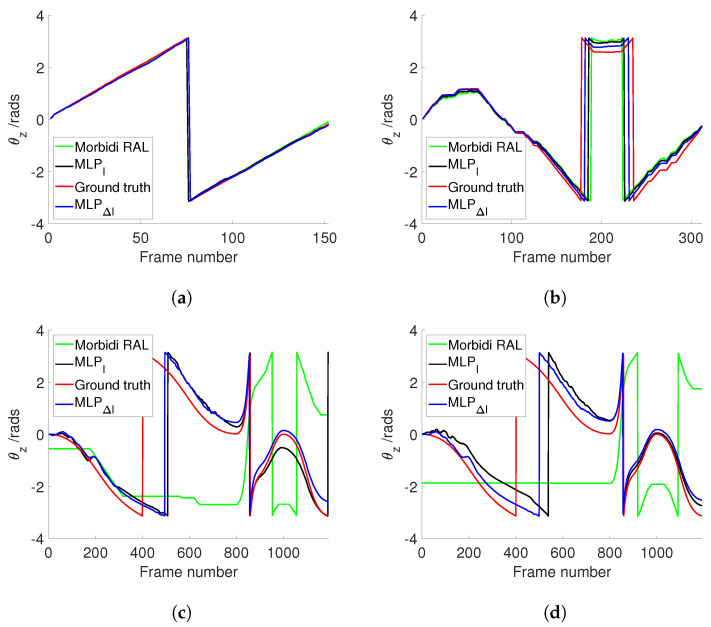
Comparison of different approaches to estimate $\theta_{z}$. (**a**) PanoraMis sequence 5. (**b**) PanoraMis sequence 6. (**c**) Synthetic dataset. (**d**) Synthetic dataset with harsh conditions.

**Figure 9 sensors-24-02246-f009:**
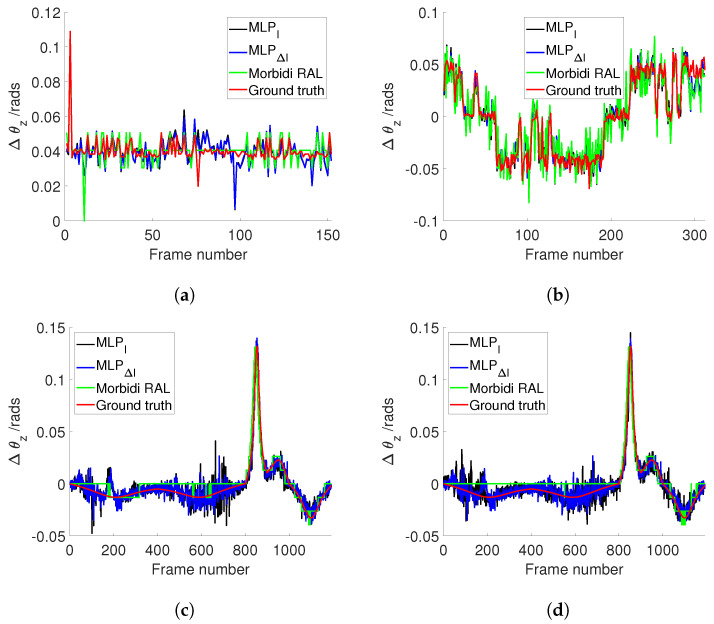
Comparison of different approaches to estimate $\Delta \theta_{z}$. (**a**) PanoraMis sequence 5. (**b**) PanoraMis sequence 6. (**c**) Synthetic dataset. (**d**) Synthetic dataset with harsh conditions.

**Figure 10 sensors-24-02246-f010:**
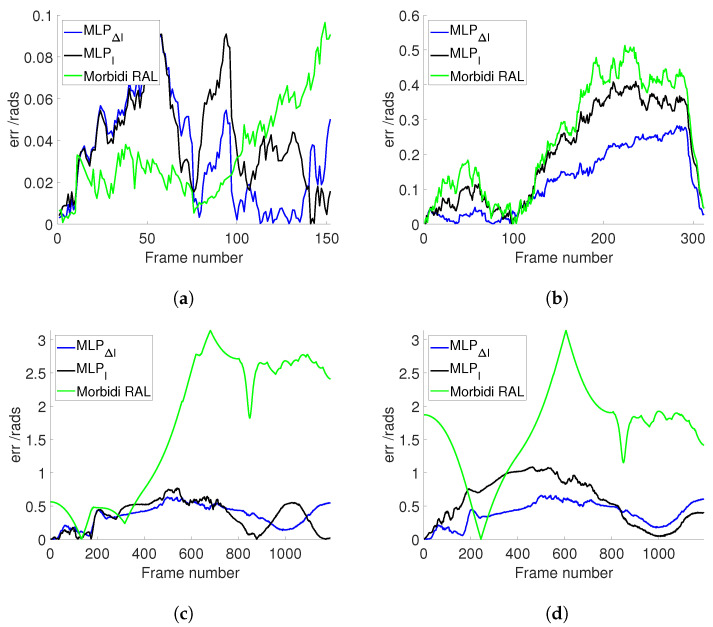
Comparison of different approaches in estimation Error. (**a**) PanoraMis sequence 5. (**b**) PanoraMis sequence 6. (**c**) Synthetic dataset. (**d**) Synthetic dataset with harsh conditions.

## Data Availability

The raw data supporting the conclusions of this article will be made available by the authors on request.

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
