# Peer review of "A Multilayer Perceptron-Based Spherical Visual Compass Using Global Features"

_sensors, 2024, doi:10.3390/s24072246_

Round 1
Reviewer 1 Report
Comments and Suggestions for Authors
The proposed approach has creativity in contribution and methodology. But, revision in terms of technical details is needed before acceptance. Also, paper organization should be improved. In this respect, some comments are suggested to describe technical details.
1. Why did you think that 16 neurons is sufficient in MLP? Discuss about it with more details.
2. Discuss about the figure 8.a with more details in a clear way. What did you report using this plot?
3. Discuss about the runtime of your proposed method (Compare with existing methods is not needed)
4. Main aim of this manuscript is MLP. So, it is suggested to review more related papers which propose new variants of MLP. For example, I find one paper titled “Developing a Tuned Three-Layer Perceptron Fed with Trained Deep Convolutional Neural Networks for Cervical Cancer Diagnosis”, which proposes new versions of MLP. Cite the paper and some related as future works and related options.
5. Discus about this phrase with more details “if the window size is 5, there are initially 4 combinations”. How do you calculate the number of combinations?
6. Discuss about hyper-parameter optimization in a clear way. Discuss about used grid exploration approach with more technical details.
Author Response
Comment and suggestions for authors
The proposed approach has creativity in contribution and methodology. But, revision in terms of technical details is needed before acceptance. Also, paper organization should be improved. In this respect, some comments are suggested to describe technical details.
Authors response
We thank the reviewer for his valuable feedback on our manuscript. We appreciate your positive comments on the creativity in contribution and methodology. We acknowledge your suggestion regarding the need for revisions in technical details and improvements in paper organization. We diligently addressed these concerns to ensure a more robust and coherent presentation of our research.
- Why did you think that 16 neurons is sufficient in MLP? Discuss about it with more details.
Authors response
The decision to use 16 neurons in the MLP was mainly influenced by the limitations of computational resources. Due to the constraints imposed by power-limited machines, selecting a smaller network architecture was crucial to ensure efficiency throughout the process of model training and deployment. This choice reflects the importance of striking a balance between model complexity and resource limitations.
We have rewritten Section 3.5.2 (from line 112 to 255) to clarify that proposing a Multilayer Perceptron (MLP) with a limited number of layers offers several advantages in various applications. Firstly, a smaller number of layers enhances computational efficiency, making the model more suitable for scenarios with limited computational resources. Additionally, a reduced number of layers helps mitigate overfitting, particularly when datasets are small, by preventing the model from capturing noise or irrelevant patterns. Smaller MLP architectures are also more interpretable, facilitating a clearer understanding of the learned relationships. Faster training speed, improved memory efficiency, and reduced susceptibility to gradient-related issues further contribute to the appeal of MLPs with a modest number of layers.
- Discuss about the figure 8.a with more details in a clear way. What did you report using this plot?  
Authors response
Figure 8.a depicts a comparison of $\theta_z$ estimation using the approach introduced in [52] and our two method variations, employing raw and gradient images as inputs. The plot is designed to assess our method's error estimation performance under conditions where there is no translation component in the motion. It is important to note that linear motion contributes to increased uncertainty in rotation estimation. This visual comparison serves as an insightful gauge of our method's consistency with previous research.
To enhance clarity, Section 4.2 has been revised (from line l.291 to l.305) in our manuscript.
Additionally, a supplementary video has been included to showcase the performance of our method.
- Discuss about the runtime of your proposed method (Compare with existing methods is not needed)
Authors response
The efficiency of our proposed method's runtime is a pivotal consideration. Rather than engaging in direct comparisons with existing methods, our emphasis lies in showcasing the efficiency of our approach, particularly highlighting the utilization of computing moments and a lightweight MLP. As indicated in response to question 1, sections 3.5 and 5, including the conclusion, have been revised to underscore this aspect.
Section 5 has been adjusted (from line l.341 to l.355) to provide clarity on this matter in the revised version.
- Main aim of this manuscript is MLP. So, it is suggested to review more related papers which propose new variants of MLP. For example, I find one paper titled “Developing a Tuned Three-Layer Perceptron Fed with Trained Deep Convolutional Neural Networks for Cervical Cancer Diagnosis”, which proposes new versions of MLP. Cite the paper and some related as future works and related options.
Authors response
We appreciate the reviewer's insightful comment, and we believe that the suggested reference has the potential to enhance related works by contributing to the state of the art in utilizing MLPs for addressing common challenges in computer vision. In the updated version of the manuscript, the reviewer can examine the modifications made between lines l.82 to l.91 in section two.
Additionally, alongside the reference recommended by the reviewer, we are also contemplating the inclusion of the following references to further support the justification for employing MLPs.:
41. Fekri-Ershad, S.; Alsaffar, M.F. Developing a tuned three-layer perceptron fed with trained deep convolutional neural networks 412
for cervical cancer diagnosis. Diagnostics 2023, 13, 686. 413
42. Yu, T.; Li, X.; Cai, Y.; Sun, M.; Li, P. S2-mlp: Spatial-shift mlp architecture for vision. In Proceedings of the Proceedings of the 414
IEEE/CVF winter conference on applications of computer vision, 2022, pp. 297–306. 415
43. Tolstikhin, I.O.; Houlsby, N.; Kolesnikov, A.; Beyer, L.; Zhai, X.; Unterthiner, T.; Yung, J.; Steiner, A.; Keysers, D.; Uszkoreit, J.; et al. 416
MLP-Mixer: An all-MLP Architecture for Vision. In Proceedings of the Advances in Neural Information Processing Systems; 417
Ranzato, M.; Beygelzimer, A.; Dauphin, Y.; Liang, P.; Vaughan, J.W., Eds. Curran Associates, Inc., 2021, Vol. 34, pp. 24261–24272. 418
44. Amjoud, A.B.; Amrouch, M. Object detection using deep learning, CNNs and vision transformers: a review. IEEE Access 2023. 419
- Discus about this phrase with more details “if the window size is 5, there are initially 4 combinations”. How do you calculate the number of combinations?
Authors response
The statement pertains to the generation of image pairs within a temporal window for our method. In this context, we were elucidating the input process of our method, comparing scenarios with and without motion augmentation. Specifically, we highlighted the sets of images introduced to our method when applying data augmentation versus when not applying it, represented as [[1,2],[1,2],[1,3],...,[1,5],[2,1],[2,3],...,[5,3],[5,4]], as opposed to [[1,2],[2,3],[3,4],[4,5]]. The calculation of combinations in this context follows the equation for combinations with repetitions, denoted as w(w-1), where w is the number of images within the temporal window. In the provided example, with a window size of 5, the number of combinations is calculated as 20.
In the revised version of the paper, enhancements have been made to the text from line l.190 to l.205 in section 3.5.1, along with the inclusion of the formulation for determining the number of possible combinations.
- Discuss about hyper-parameter optimization in a clear way. Discuss about used grid exploration approach with more technical details.  
Authors response
We concur with the reviewer's suggestion that this aspect requires more detailed clarification. Hyper-parameter optimization is a pivotal step in refining our model architecture. Our methodology entails a meticulous evaluation of the model's performance across varying numbers of layers (depth) and neurons per layer (width), the hyper-parameters under scrutiny, employing the grid search/exploration algorithm. Consequently, we have refined the original sentence, "We used a grid exploration approach to tune different hyper-parameters (depths and widths)," in the revised version of the text from line l.180 to l.186 of section 3.5.
Reviewer 2 Report
Comments and Suggestions for Authors
The authors present a method that includes masks and a mlp for a visual compass task. The proposed method is incremental and is based on spherical moments. It is an important task in robotics.
The structure of the paper could be reviewed as some experimental results are shown in 3.5.
Some important details are missing about the experiments. For example, the total number of images used for training is not clear, or the proportion between simulated and real data. The authors claimed that their method is efficient but do not provide any information about processing time or memory load.
The sota methods used for comparison are outdated and it is not clear why the authors used [51] for their experiments while claiming that their method is more efficient than [49].
Results are also only shown for 2 sequences from the panoramis dataset with no mention of the others. Qualitative results could also be shown in images.
simulated data does not seem realistic as the major part of the images are sky views which are not useful in a practical setup.
References are adequate.
Comments on the Quality of English Language
The paper is easy to read but there are still some typos in the text (l. 102, 134, 135,…). Fig 1 does not show the output of the mlp.
Author Response
Comment and suggestions for authors
The authors present a method that includes masks and a mlp for a visual compass task. The proposed method is incremental and is based on spherical moments. It is an important task in robotics. 
Authors response
We express our sincere gratitude for your thoughtful review of our manuscript. Your positive recognition of our method, which integrates masks and employs an MLP to tackle the visual compass estimation task using spherical moments, is truly motivating.
The structure of the paper could be reviewed as some experimental results are shown in 3.5.
Authors response
We concur with the reviewer's suggestion to revise the paper structure, particularly in sections 3.5 and 4. In the original version, some experimental analysis was present in the methodology section (Section 3). Following this advice, we have restructured the paper by eliminating Section 3.5.3 and integrating its content into Section 4 (experiments), as it was more aligned with a discussion of experimental setup rather than methodology. We invite the reviewer to review the updated manuscript to examine the modifications in sections 3.5 and 4.
Some important details are missing about the experiments. For example, the total number of images used for training is not clear, or the proportion between simulated and real data.
Authors response
We appreciate this observation and have incorporated the mentioned information into the updated version of the paper, specifically in the new 4.1 section titled "Data preparation." This section outlines that out of a total of 2792 images, 474 images were sourced from the real-world dataset Panoramis, constituting approximately 17% of real images in the dataset. We invite the reviewer to review the updated manuscript between lines 255 and 259.
The authors claimed that their method is efficient but do not provide any information about processing time or memory load. 
Authors response
We express our gratitude to the reviewer for this feedback. As previously detailed in the original version of the paper (line 162), the computational cost for computing moments with a single mask, following Equation (9), is approximately 10 microseconds (10^-5 sec) in the Matlab implementation. Additionally, considering the reliance on the Procrustes method (with a residual impact on computation cost) and the proposed MLP (with a response time also around 10 microseconds), the overall computation cost for the proposed utilization of 100 masks results in a method achieving a performance of around 1 millisecond per image.
To enhance clarity on this aspect, we have refined the conclusions in Section 5. Specifically, modifications have been made to lines 346 to 352 in the revised manuscript.
The sota methods used for comparison are outdated and it is not clear why the authors used [51] for their experiments while claiming that their method is more efficient than [49]. 
Authors response
Indeed, the approach [49] (revised version [53]) stems from a prior work by one of the authors of this manuscript, which primarily focused on the visual servoing aspect rather than feature extraction and pose estimation. In contrast, the reference [51] (revised as [55]) is more apt for comparison since the work by Morbidi et al. also addresses the challenge of rotation estimation.
Results are also only shown for 2 sequences from the panoramis dataset with no mention of the others. Qualitative results could also be shown in images. 
Authors response
We acknowledge this feedback.
We have used just sequence 5 and 6 because we present this method focusing on this specific sort of lens. We focus on this kind of lens because of its wide angle.
To provide a more comprehensive illustration of the performance of our method across various datasets used in the experiments, we have incorporated a video showcasing additional qualitative results. The link to this supplementary material is included within the manuscript.
simulated data does not seem realistic as the major part of the images are sky views which are not useful in a practical setup. 
Authors response
As pointed out by the reviewer, simulation data may lack realism and can be challenging to generalize to real-world scenarios. In response, we adopted a hybrid approach, incorporating both simulated and real-world datasets.
Concerning the viewpoint of the virtual camera in the canyon dataset, we opted for consistency with the Panoramis dataset, which features a catadioptric camera pointing upward.
References are adequate. 
Authors response
We thank the reviewer for this comment.
Comments on the Quality of English Language
The paper is easy to read but there are still some typos in the text (l. 102, 134, 135,…). Fig 1 does not show the output of the mlp. 
Authors response
We would like again to express our gratitude to the reviewer for providing valuable feedback. In response to the insightful comments, we have addressed the detected errors in our revised version. Special attention has been given to rectify typos. In addition, we have made modifications to Figure 1 and its caption to provide a clearer explanation of the output of the MLP, denoted as \(R_{MLP} \in SO(2)\).
Reviewer 3 Report
Comments and Suggestions for Authors
The paper introduces a novel method for a visual compass using global features, specifically spherical moments. By combining image masks, neural networks, and other techniques, the proposed approach offers a unique solution handling changes due to camera translation motion and scene depth discontinuities. Authors also claim real-time inference speed, making it suitable for embedded robot applications.
A novel contribution of the proposed method is the use of image masks to avoid abrupt changes in global feature values caused by the appearance and disappearance of regions in the camera's field of view as it moves.
A visual compass system can theoretically estimate these three rotational degrees of freedom enabling determination of the camera's orientation based on the captured spherical images, but only one orientation angle (vehicle moving in a plane) is presented in the comparative tables, so it does not incorporate the estimation of the pitch that can produce the displacement in an uneven terrain.
The algorithm is evaluated with both public databases and a database of synthetic omnidirectional images created specifically by the authors. The paper presents innovative data augmentation strategies, such as the shift folder-window strategy, to enhance the training dataset.
Overall, the proposed architecture presents relevant contributions. Format of the paper and English grammar are adequate. The manuscript is well organized and redacted. The mathematical background and references are adequate, the proposed method is fairly well described and the research methodology adequate. Bibliography is extensive and up-to-date.
Comments on the Quality of English LanguageFormat of the paper and English grammar are adequate
There are some minor grammar and vocabulary errors in the paper that need to be corrected (see comments below):
Line 1: “ This paper proposes visual compass method” misses the article: “a visual compass method”
Line 5: “is also a big challenge since it depends on the scene depths. Especially for non-planar scene.” This is just a single sentence, replace the period by a comma
Author Response
Comment and suggestions for authors
The paper introduces a novel method for a visual compass using global features, specifically spherical moments. By combining image masks, neural networks, and other techniques, the proposed approach offers a unique solution handling changes due to camera translation motion and scene depth discontinuities. Authors also claim real-time inference speed, making it suitable for embedded robot applications.
A novel contribution of the proposed method is the use of  image masks to avoid abrupt changes in global feature values caused by the appearance and disappearance of regions in the camera's field of view as it moves.
A visual compass system can theoretically estimate these three rotational degrees of freedom enabling determination of the camera's orientation based on the captured spherical images,  but only one orientation angle (vehicle moving in a plane) is presented in the comparative tables, so it does not incorporate the estimation of the pitch that can produce the displacement in an uneven terrain. 
The algorithm is evaluated with both public databases and a database of synthetic omnidirectional images created specifically by the authors. The paper presents innovative data augmentation strategies, such as the shift folder-window strategy, to enhance the training dataset.
Overall, the proposed architecture presents relevant contributions. Format of the paper and English grammar are adequate. The manuscript is well organized and redacted. The mathematical background and references are adequate, the proposed method is fairly well described  and the research methodology adequate. Bibliography is extensive and up-to-date. 
Authors response
Thank you sincerely for your thoughtful and comprehensive review of our manuscript. We appreciate your positive comments regarding the novel contributions of our proposed method for visual compass, incorporating global features like spherical moments, image masks, and neural networks.
Your insightful observation regarding the limitation in presenting only one orientation angle in the comparative tables is duly noted. We understand the importance of considering all three rotational degrees of freedom, especially in scenarios involving uneven terrains. In fact, we are currently working on the proposal of a visual gyroscope (3DoF) that extends the functionalities of this proposal, a visual compas (1DoF). But what corresponds to this manuscript, in our revised version, we ensured to include a more comprehensive analysis (by upgrading section 3 and 4 and improving conclusion). Also, we believe that the incorporation of more qualitative results, like the supplementary material, and a video showing the performance of our method in different conditions helps to address the concerns of the reviewer.
We are pleased that you found our evaluation strategy with both public and synthetic databases commendable. Your recognition of the innovative data augmentation strategies, such as the shift folder-window strategy, is appreciated. We strived to enhance the clarity of these strategies in the revised manuscript.
Your positive feedback on the overall structure, English grammar, mathematical background, and references is appreciable.
Thank you once again for your time and valuable insights.
Comments on the Quality of English Language
Format of the paper and English grammar are adequate
 
There are some minor grammar and vocabulary errors in the paper that need to be corrected (see comments below):
 
Line 1: “ This paper proposes  visual  compass method” misses the article: “a visual compass method”
Line 5: “is also a big challenge since it depends on the scene depths. Especially for non-planar scene.” This is just a single sentence, replace the period by a comma
Authors response
We acknowledge again the reviewer for this comment. In our revised version, we have corrected the detected errors.
Round 2
Reviewer 1 Report
Comments and Suggestions for Authors
Revised version is better than original submission in terms of paper organization and technical details. Most of comments have been considered in this version. The proposed method is decribed in a more clear way.
Author Response
Thank you for taking the time to review our revised manuscript. We appreciate your positive feedback regarding the improvements made in the revised version, particularly in terms of paper organization and clarity of technical details.
We have made efforts to enhance the clarity and coherence of the proposed method, ensuring that it is described in a more explicit and understandable manner.
In this second revision, we have corrected some typos and rephrase parts of the text in order to give more clarity to the manuscript. The modifications were highlighted in blue.
Reviewer 2 Report
Comments and Suggestions for Authors
The authors have addressed the issues from the previous reviews.However, the paper still needs to be proofread as there are still many typos.
Comments on the Quality of English LanguageThere are still many typos in the paper e.g. e.g. l.303, 257, 250, 237,194, 203...
Author Response
Thank you for your valuable feedback on our manuscript. We have carefully considered your observation regarding the presence of typos in the manuscript despite our efforts to address previous concerns raised by the reviewers. We acknowledge the importance of ensuring the clarity and correctness of the manuscript.
In response to your feedback, we have realized another round of comprehensive proofreading to rectify any remaining typographical errors and enhance the overall readability of the manuscript.
The new version of the paper highlights in blue the modifications (specially, methodology and experimental sections) with respect the previous version.